# Metallic On-Chip Light Concentrators Fabricated by In Situ Plasmonic Etching Technique

**DOI:** 10.3390/nano12234195

**Published:** 2022-11-25

**Authors:** Lihua Cha, Pan Li

**Affiliations:** 1School of Law, Central University of Finance and Economics, Beijing 100081, China; 2Institute of Analysis and Testing, Beijing Academy of Science and Technology, Beijing 100089, China; 3Department of Physics, Capital Normal University, Beijing 100048, China; 4School of Information Technology, Beijing City University, Beijing 100083, China

**Keywords:** surface plasmons, nanowaveguide, nanofocusing, metallic nanowire

## Abstract

One-dimensional tapered metallic nanostructures are highly interesting for nanophotonic applications because of their plasmonic waveguiding and field-focusing properties. Here, we developed an in situ etching technique for unique tapered crystallized silver nanowire fabrication. Under the focused laser spot, plasmon-induced charge separation of chemically synthesized nanowires is excited, which triggers the uniaxial etching of silver nanowires along the radial direction with decreasing rate, forming tapered structures several micrometers long and with diameter attenuating from hundreds to tens of nanometers. These tapered metallic nanowires have smooth surfaces showing excellent performance for plasmonic waveguiding, and can be good candidates for nanocircuits and remote-excitation sources.

## 1. Introduction

Plasmonic nanowaveguides possess the unique ability to guide and manipulate light at subwavelength confinement [1,2]. Control over the plasmon propagation is critical in order to construct plasmonic nanocircuits and nanodevices such as plasmon modulators [3,4], routers [5] and sensors [6,7,8], etc. In particular, the morphology of waveguides can modulate the property of propagating surface plasmons (SPs) [9]. It is well known that tapered metallic nanowaveguides can channel and focus plasmons to generate enormous near fields, known as nanofocusing [10,11,12]. Nanofocusing tapered nanostructures are effective tools to tightly focus light waves and to avoid electromagnetic field radiation into the free space. However, current approaches to constructing such plasmonic nanostructures have commonly relied on sophisticated fabrication techniques, such as electron beam lithography and focused ion beam milling. This inevitably introduces inherent surface roughness that substantially degrades the optical performance [13].

Plasmon-induced charge separation (PICS) is a process of energy nonradiative transition, in which free electrons transfer from a metal surface to a contacted semiconductor by absorbing plasmon energy and lead morphological changes of the metallic nanostructures [14]. It has been widely reported that the PICS process can be applied to photochromics [15,16], nanoimaging [17] and photocatalysis [18], etc. Meanwhile, the potential of PICS as a promising on-chip manufacturing method for SPs functional waveguide fabrication is rarely reported [19,20].

In this paper, we propose that the PICS effect can be utilized to achieve Ag nanowires (NWs) etching. By focusing a continuous wave laser on chemical synthesized NWs, tapered nanostructures with smooth surfaces and diameter attenuating from hundreds to tens of nanometers can be formed within an illuminating time of tens of seconds. Verified by the observation of non-uniform SP beats, these tapered NWs are excellent nanofocusing waveguides with divergent effective refractive indices for plasmon modes. These findings demonstrate a promising possibility for on-chip manufacture of functional nanostructures through the PICS method by designing the optical field distribution.

## 2. Experimental Setup

Plasmonic etching of the Ag NWs: The chemically synthesized NWs (see Appendix A for details of the used NWs) were first dropped onto the substrate (ITO glass coated with 20 nm TiO_2_) and submerged in deionized water. A 633 nm CW laser with power in the order of milliwatts was focused onto the NW for the etching. Here, the objective (50×) with numerical aperture 0.45 was used, forming a Gaussian distributed spot with effective excitation area about 200 μm^2^. The etching process was detected in situ through a dark field microscopy.

Mapping the SPs propagation in the etched NWs: Propagation of the SPs was launched by focusing a laser beam (633 nm) through an oil immersed objective (100 ×/NA = 1.3) on the thicker end of the wire. To visualize the near field distributions of surface plasmons, the NWs were first covered with a layer of aluminum oxide (Al_2_O_3_) with a thickness of 10 nm by atomic layer deposition to prevent quenching of the QDs fluorescence. Then, CdSe/ZnS core/shell quantum dots (QDs, Invitrogen, SKU# Q21321MP, diluted 200 times) were spin-coated at a speed of 400 r/min onto the sample. QDs adjacent to the NW can be excited by evanescent fields of propagating plasmons that are launched by focusing the laser at the thicker end of the NW. These luminescent QDs act as “reporters” of the local field at every point along the entire structure. After filtering the laser light, the fluorescence image was recorded by the CCD detector.

Simulations of the SPs propagating on the etched NWs: Electromagnetic calculations were based on a finite element method (COMSOL Multiphysics). The model is a tapered wire with a pentagonal cross-section on a TiO_2_ coated ITO substrate covered by a layer of 10 nm Al_2_O_3_ surrounded by oil. The field distributions and effective refractive indices of eigenmodes were obtained using the mode solver in COMSOL. The dielectric permittivity of silver is taken from Johnson and Christy [21]. The refractive indices used for TiO_2_, ITO, Al_2_O_3_ and oil are 2.55, 1.78, 1.76 and 1.53, respectively.

## 3. Results and Discussion

The on-chip etching requires semiconductor substrate to connect the metallic nanostructures, so that the plasmon-induced separating electrons of the NW can transfer to the substrate according to the schematic shown in Figure 1a. In the experiments, we used TiO_2_ coated ITO glass as the substrate, while the NWs were immersed in deionized water to facilitate the etching process. To perform the etching, a Gaussian distributed laser spot is focused onto the NW. During the laser illumination, the localized plasmons on the NW trigger hot electron generation. Once the hot electrons’ energy reaches the conduction band of the semiconductor substrate, a hot current toward the substrate can be formed as the electrons of the NW separate and transfer, while the metal atoms on the NW are oxidized to ions and therefore dissolve into the water. Because the energy of the TiO_2_ conduction band is very close to the Fermi level of metal Ag, the etching can happen in a very short time under excitation of the 633 nm laser [17]. To detect and control this process in situ in real time, we use a homebuilt dark field microscope system to perform the experiments. The dark field images of a typical NW after 60 s etching is shown in Figure 1b, where the laser with polarization perpendicular to the NW long axis is focused on the middle of the NW through the objective of the microscope. We found a significant morphological change of the NW with only 30 s of illumination time, as shown in Figure 1b(ii). After 60 s etching, the NW was fused to two separated nanoneedles, and the nanostructure within the laser spot was not found completely. The power of the laser used in the etching was 200 mW. Besides TiO_2_, we also perform NW etching with 20 nm ZnO coated ITO substrate, with similar results (see Appendix A). We emphasize that the PICS is a general effect only requiring that the energy of hot electrons reaches the conduction band of the substrate, which is promising to utilize in different on-chip fabricating environments with various substrates. To exclude laser ablation effects on the morphological changes of the NWs, the experiment was performed on glass substrate with other conditions unchanged. However, after 200 s illumination, no significant change of the NW could be observed, as shown in Appendix A. These results confirm that the semiconductor substrate is the key to the NW etching.

To quantify the relationship between the NW morphological changes and the laser illuminations, we measured the intensity distribution of the laser spot on the NWs. Figure 2a shows a dark field image of an Ag NW after 25 s laser illumination. The region of the NW within the laser spot is marked as A to B, while the inset shows the according cross-line of the laser intensity distribution on the NW as detected by the dark field microscopy camera. The NW morphology in Figure 2a was detected by SEM. The images are shown in Figure 2b,c. This demonstrates that the NW outside the illuminating area has a diameter of around 110 nm. In the illuminating area, the NW formed a tapered structure with a smooth surface, while the diameter decreased from 110 nm to 50 nm. Then, we measured the diameter changes at different positions of the NW, and compared with the according laser intensity profile. The result, as shown in Figure 2d, suggests that the diameter changes have a linear correspondence with the laser intensity. Figure 2e shows diameter changes of more NWs etched under the same experiment conditions, and the results are fitted using a linear function. It indicates that this method could be utilized as a general in situ on-chip etching technique for metallic nanostructures’ design and fabrication by contriving light field distribution.

To unveil the mechanism of the on-chip etching, the excitation polarizations of the laser were also investigated. Figure 3 shows an Ag NW exposure to a laser with perpendicular and parallel polarizations at different positions. The laser power was fixed at 200 mW. After 25 s, the NW morphology showed a significant change for the perpendicular excitation area, while the parallel excitation area remained unchanged. This suggests that the etching is a polarization-dependent process. It is reported that the near field intensity on the NW surface is highly dependent on the excitation polarization [22]. While the NW is under a perpendicular excitation, localized SPs are generated, and thus a drastically enhanced field is excited at the surface of the NW. On the other hand, the parallel excitation induces much lower near field intensity due to the light scattering effect of the NW [22]. This result indicates that the localized SPs on the NW play a major role in the etching process.

Because the tapered NWs preserve smooth surfaces during on-chip etching, they can serve as quality nanofocusing waveguides. In light of this idea, we researched the SPs propagation on the etched NWs. Figure 4a shows a dark field image of a typical etched NW. The SPs were excited by a focused laser on the end of the NW with parallel polarization. To reveal the SPs propagating, we visualized the near field distribution along the NW by quantum dots based local-field imaging. As shown in Figure 4b, a series of quasiperiodic node-like field distributions along the NW can be observed. This near field spatial modulation can originate from the superposition of two propagating modes H_0_ and H_2_. For a longitudinal excitation, only these two modes exist on the Ag NW, where H_0_ is a longitudinal mode with electrons oscillating parallel to the wire axis, and H_2_ is a transverse mode with electrons oscillating vertically to the substrate [23]. The field distributions of the two modes are shown in Figure 4c, where the H_0_ mode is mainly confined at the interface of the NW and the substrate, and the H_2_ mode with higher intensity mainly around the top surface of the NW. The length of the node, *Λ*, resulting from the interference of these two modes is determined by the equation: *Λ* = λ_0_/(*n*_H0_ − *n*_H2_), where *λ*_0_ is the wavelength in vacuum, and *n*_H0_, *n*_H2_ are the effective refractive indices for the two modes, respectively. We measured the node length along the longitudinal axis and found it changed from 1.1 µm to 0.5 µm, as shown in Figure 4d. This decrease can be quantitatively evaluated by the relationship of the effective refractive indices to the diameter. As shown in Figure 4e, the *n*_H0_ increases rapidly with decreasing diameter, while the *n*_H2_ remains almost constant. The result of the increasing *n*_H0_ implies that there is a plasmon nanofocusing effect of the NW [24], which generates a high localization and enhancement of the near field around the thinnest diameter of the NW. It also indicates that the on-chip etching process preserves the plasmonic quality of the nanostructures.

## 4. Conclusions

In summary, we have reported a novel approach to realizing an in situ etching technique for Ag NWs and achieving fabrication of tapered NWs. By changing the excitation polarizations, we show that the morphological changes of the NWs are mainly induced by the plasmonic enhanced near field on the NWs. The tapered morphology is the consequence of the Gaussian excitation created by the focused light field. These fabricated tapered NWs can guide SPs over a considerable distance and adiabatically compress the optical field to thinner positions, and thus can be a good candidate for waveguiding and remote-excitation sources. As for the fabrication method, we have showed that the light field distribution can effectively tailor the morphology of the chemically synthesized NWs. These results also illustrate the methodology for controlled etching of plasmonic nanostructures by designed optical field distribution.

## Figures and Tables

**Figure 1 nanomaterials-12-04195-f001:**
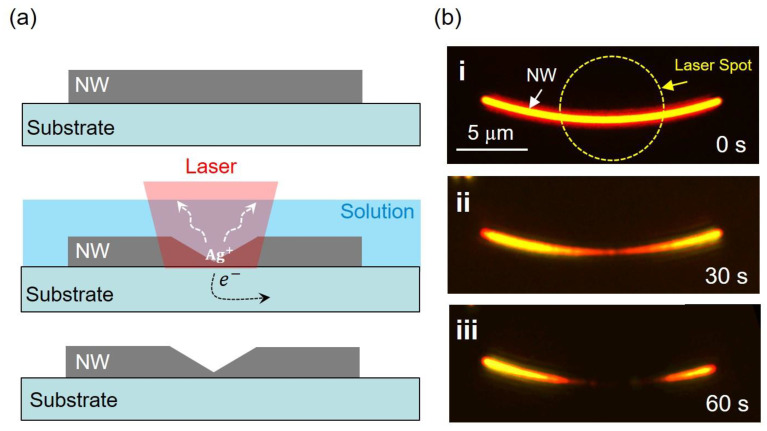
On-chip etching of Ag NW. (**a**) Schematic images of NW etching. (**b**) Dark field images of the etching process with illumination time (**i**) 0 s; (**ii**) 30 s; (**iii**) 60 s.

**Figure 2 nanomaterials-12-04195-f002:**
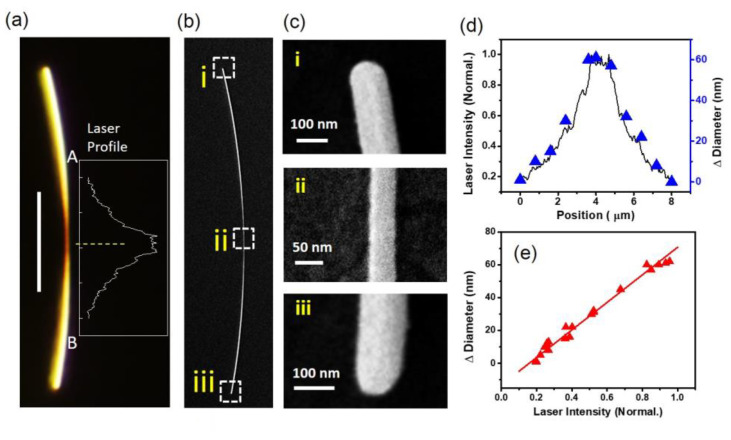
Relationship between NW morphological changes and laser intensity. (**a**) Dark field image of a NW after 25 s laser illumination. The laser power is 200 mW. The white dashed circle indicates the focused laser spot. The scale bar is 5 µm. (**b**) SEM image of the NW in (**a**). (**c**) Zoomed in SEM images of the area (**i**–**iii**). (**d**) Laser intensity and NW diameter changes from position A to B along the NW as shown in (**c**). (**e**) Relationship between NW diameter changes and laser intensity.

**Figure 3 nanomaterials-12-04195-f003:**
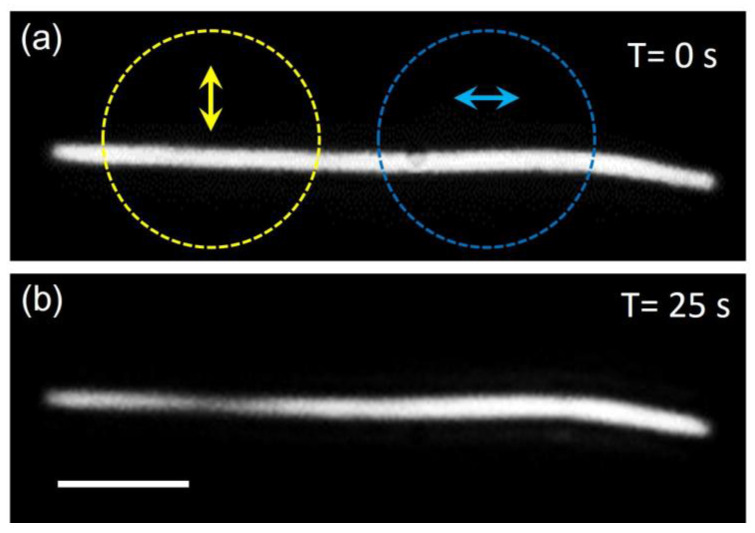
NW etching with different excitation polarizations. (**a**) Dark field image of an Ag NW. The dashed circles indicate laser spots with polarization of perpendicular (yellow), and parallel (blue). (**b**) Dark field image of the NW in (**a**) after etching. The scale bar is 5 μm. The illumination time is 25 s.

**Figure 4 nanomaterials-12-04195-f004:**
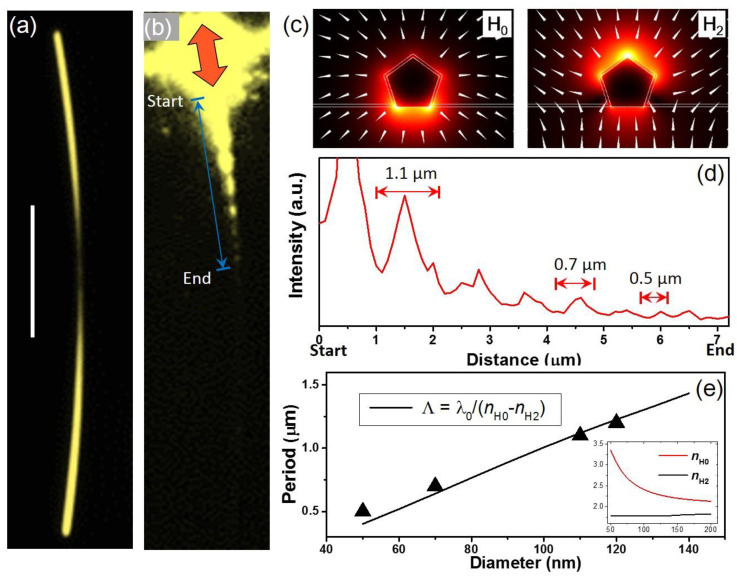
Plasmons propagate on an etched NW. (**a**) Dark field image of an etched NW. The scale bar is 5 µm. (**b**) QDs emission image of the plasmon propagating along the NW. The wavelength of the excitation laser is 633 nm. The red arrow indicates the polarization of the excitation. (**c**) Calculated electric field distributions of the two plasmon modes H_0_ and H_2_ for a NW with diameter of 100 nm under the excitation of 633 nm laser. White arrows mark the directions of the electric field. (**d**) The emission intensity profile along the NW. (**e**) Beat length as a function of the NW diameter. The black curve represents the calculated result and the triangular dots represent the experimental data for each observed beat length. The inset shows calculated effective refractive indices, *n*_H0_ and *n*_H2_, for the H_0_ and H_2_ modes as a function of the diameters.

## Data Availability

Not applicable.

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
