# Peer review of "Metallic On-Chip Light Concentrators Fabricated by In Situ Plasmonic Etching Technique"

_nanomaterials, 2022, doi:10.3390/nano12234195_

Round 1

Reviewer 1 Report

The research is scientifically sound, the topic is interesting and its relevance and the context in the field is clear.

The main question is related to "the ability to guide and manipulate light at nanoscale confinement". The research deals with the potential of PICS as a promising on-chip manufacturing method for SPs functional waveguide fabricating

The research is scientifically sound, the topic is interesting and it is relevant in the field. The authors presented and discussed the potential of PICS as a promising on-chip manufacturing method for the Ag nanowires (NWs) etching as a novel technique. 

The novelty and the advance was explained in the introduction. The PICS process was applied to photo-chromics [15, 16], nanoimaging [17] and photocatalysis [18], etc but only few works about on-chip manufacturing method. The authors has to include these “only few works” within the reference list.

The conclusions are consistent with the evidence and arguments presented and do they address the main question posed

The authors claimed that there are only only few works on the potential of PICS as a promising on-chip manufacturing method for SPs functional waveguide fabricating. The authors has to include these “only few works” within the reference list.

The tables and figures are clear and well presented.

Author Response

Thanks the referee for the positive comments and constructive suggestions to the paper. We have revised the manuscript accordingly. Please see the attachment for the point-by-point response.

Reviewer 2 Report

L. Cha and P. Li reported in-situ plasmonics etching technique to fabricate metallic on-chip light concentrator using Ag Nanowire(NW). I think this technique is interesting to readers of this journal. However, the following issues should be addressed to be published in the journal.

1. It would be better for the authors to represent the simulated results of the field distribution for the two propagation modes (H0, H1) within the NW in the main text. Then, it is also necessary to describe the difference in field intensity between the Ho and H1 modes.

2. The authors carried out experiments after forming Ag NWs on ITO substrates coated with 20-nm thick TiO2. Is there a reason why the 20-nm thick TiO2 layer was chosen here? It would be good to include an additional explanation of whether similar results would be obtained if run on other semiconductor materials.

3. The authors should appropriately improve the English expression in parts of the main text.

Author Response

(The authors gave the same response as above.)
